# Potential of Rapid Tooling in Rapid Heat Cycle Molding: A Review

**DOI:** 10.3390/ma15103725

**Published:** 2022-05-23

**Authors:** Nurul Hidayah Mohamad Huzaim, Shayfull Zamree Abd Rahim, Luqman Musa, Abdellah El-hadj Abdellah, Mohd Mustafa Al Bakri Abdullah, Allan Rennie, Rozyanti Rahman, Sebastian Garus, Katarzyna Błoch, Andrei Victor Sandu, Petrica Vizureanu, Marcin Nabiałek

**Affiliations:** 1Faculty of Mechanical Engineering and Technology, Universiti Malaysia Perlis, Arau 02600, Malaysia; hidayahhuzaim@gmail.com; 2Center of Excellence Geopolymer and Green Technology (CEGeoGTech), Universiti Malaysia Perlis, Kangar 01000, Malaysia; luqman@unimap.edu.my (L.M.); mustafa_albakri@unimap.edu.my (M.M.A.B.A.); rozyanty@unimap.edu.my (R.R.); 3Faculty of Chemical Engineering and Technology, Universiti Malaysia Perlis, Kangar 01000, Malaysia; 4Laboratory of Mechanics, Physics and Mathematical Modelling (LMP2M), University of Medea, Medea 26000, Algeria; lmp2m_cum@yahoo.fr; 5Lancaster Product Development Unit, Engineering Department, Lancaster University, Lancaster LA1 4YW, UK; a.rennie@lancaster.ac.uk; 6Faculty of Mechanical Engineering and Computer Science, Częstochowa Univerity of Technology, 42-201 Częstochowa, Poland; sebastian.garus@pcz.pl; 7Department of Physics, Częstochowa University of Technology, 42-201 Częstochowa, Poland; katarzyna.bloch@pcz.pl (K.B.); nmarcell@wp.pl (M.N.); 8Faculty of Materials Science and Engineering, Gheorghe Asachi Technical University of Iasi, 41 D. Mangeron Street, 700050 Iasi, Romania; sav@tuiasi.ro; 9Romanian Inventors Forum, Str. Sf. P. Movila 3, 700089 Iasi, Romania; 10Technical Sciences Academy of Romania, Dacia Blvd 26, 030167 Bucharest, Romania

**Keywords:** rapid tooling, rapid heat cycle molding, additive manufacturing, injection molding process

## Abstract

Rapid tooling (RT) and additive manufacturing (AM) are currently being used in several parts of industry, particularly in the development of new products. The demand for timely deliveries of low-cost products in a variety of geometrical patterns is continuing to increase year by year. Increased demand for low-cost materials and tooling, including RT, is driving the demand for plastic and rubber products, along with engineering and product manufacturers. The development of AM and RT technologies has led to significant improvements in the technologies, especially in testing performance for newly developed products prior to the fabrication of hard tooling and low-volume production. On the other hand, the rapid heating cycle molding (RHCM) injection method can be implemented to overcome product surface defects generated by conventional injection molding (CIM), since the surface gloss of the parts is significantly improved, and surface marks such as flow marks and weld marks are eliminated. The most important RHCM technique is rapid heating and cooling of the cavity surface, which somewhat improves part quality while also maximizing production efficiencies. RT is not just about making molds quickly; it also improves molding productivity. Therefore, as RT can also be used to produce products with low-volume production, there is a good potential to explore RHCM in RT. This paper reviews the implementation of RHCM in the molding industry, which has been well established and undergone improvement on the basis of different heating technologies. Lastly, this review also introduces future research opportunities regarding the potential of RT in the RHCM technique.

## 1. Introduction

Time to market is a critical aspect of product development strategies [1,2,3]. In general, speed is balanced with other factors such as functionality, innovation, or quality [4,5]. Today, with the emergence of various new technologies, global competition in product development is developing rapidly. In addition, various industries are constantly in search for the newest competitive technologies in order to meet rising demand, reduce costs, and be able to produce products in small batches, while ensuring high quality and the capability to reach sustainable development goals [6,7,8]. The rapid growth of RT technology in today’s global market requires rapid product development to replace conventional methods in order to improve manufacturing processes, particularly for tooling fabrication [7,8,9]. RT offers a faster manufacturing speed that reduces cost and saves project time, which is critical for successfully completing testing and moving into full production, as illustrated in Figure 1 [10].

Regardless of the size of the industry, there will be times when immediate tooling is required to address certain issues. Moreover, an intermediate tooling system is required while producing a small quantity of functional prototypes to test the performance of the product designed. Such a small volume of product is generally used as a market sample, evaluation requirement, and production process design [11,12]. Although 3D printing technologies are available to produce the prototype, the properties of the parts produced are far from the properties of the parts produced using actual manufacturing processes for mass production. Therefore, RT is definitely required in order to evaluate the actual performance of the product designed, especially related to the material properties.

For any scientific research, functional tools or prototypes need to be launched prior to mass production. These are not launched in a mass volume for the consumers but in a small amount for the researchers [11,13]. RT is extremely beneficial for this situation, as it allows the launching of the products in a short time. Furthermore, since the manufacturing cost is low, users can get the products at a low price compared to the prices of those mass-produced using production tooling. This is why several new start-up or even big companies prefer using this tool to expand their revenue and achieve a competitive edge over fellow companies in the market.

Normally, mold-making or prototyping industries use mild steel or aluminum as material for the mold inserts in RT [14,15,16]. However, the use of these types of materials requires the same machining processes used to manufacture the production tooling such as computer numerical control (CNC), electrical discharge machining (EDM), and wire EDM machining, which are costly and time-consuming [6,17]. Recently, additive manufacturing (AM) was used to fabricate mold inserts for RT [9,18,19,20,21]. Typically, RT uses a model or prototype fabricated by AM technology as a pattern to form mold inserts, or mold inserts are directly manufactured through the AM method for a minimum number of prototypes [8,22]. In industry, there are variety of RT technologies available, such as the combined process of RT and AM to lessen the time for making the RT.

According to Huang et al. [23], RT differs from conventional tooling because the time required to manufacture tools is much quicker and it can be categorized as indirect or direct processes. Direct tooling techniques use AM processes to directly manufacture mold inserts without the need for a master pattern in the process [6,24,25,26]. The processes involved in direct tooling are selective laser sintering (SLS), direct metal laser sintering (DMLS), fused deposition modeling (FDM), stereolithography (SLA), jetted photopolymer (PolyJet), and 3D printing (3D-P) [6,8,27,28]. On the other hand, indirect processes use AM parts as the master pattern to make a mold for castings or plastic molding processes. These processes include cast metal tooling, silicone molding, spray metal tooling, and 3D KelTool tooling [6,8,9,18,29].

Generally, mold inserts for RT produced using AM technologies such as SLS, FDM, SLA, PolyJet, and 3D-P exhibit low thermal and mechanical properties; therefore, they are prone to failure throughout the molding process [30,31,32]. On the other hand, the DMLS process is not suitable for a simple plastic part design as it is time-consuming compared to machining processes. Moreover, for complex parts, DMLS has a few limitations for certain feature designs. In addition, past research on RT mold inserts discovered that the stress implemented in the mold inserts throughout the injection molding cycle has a significant impact on mold life [6,17,33,34,35].

On the other hand, a new technology called rapid heat cycle molding (RHCM) has been proven to eliminate weld line and molding defects in CIM, as well as the high glossy surface, by heating the core and/or cavity insert beyond the glass transition temperature of the plastic resin [36,37,38]. Thus, by raising the mold temperature to the glass transition temperature, weld lines can be considerably eliminated [36,39,40,41,42].

Table 1 shows various types of RCHM technologies investigated by researchers to improve weld line strength, tensile strength, and surface quality using various types of RHCM technology such as electromagnetic induction heating, induction heating, hot oil, a combination of hot oil and induction heating, steam heating (vapor chamber), and electric heating in hard/production tooling.

Tosello et al. [51] investigated the impact of injection molding parameters on weld line formation and strength reduction. According to the studies, the most important factor to reduce weld lines is increasing the mold temperature. Wang et al. [52] analyzed the heat transfer during heating and cooling phases of RHCM process. The results of production showed that the RHCM process completely eliminates the weld lines on the surface of the parts, and the surface gloss of the parts is also greatly improved. The influence of the RHCM process on part surface quality has been studied [40,42,50,53,54]. The results showed that the high cavity surface temperature near to the polymer’s glass transition temperature can assist in reducing surface roughness, significantly enhancing surface gloss, and eliminating the weld line. Studies have shown that, by using the RHCM process to eliminate weld lines, the surface roughness of molded parts can be significantly reduced. As a result, processes such as spraying and coating can be eliminated, reducing energy, material, and production costs while also protecting the environment [55,56,57,58]. Thus, it is being considered as a possible manufacturing technology in the plastic injection molding industry.

Therefore, this paper intends to provide a brief review on the role of RT technology in manufacturing methods of tooling toward its potential in the RHCM technique. The review of the literature included previous studies that were conducted on the application of RHCM in the injection molding process, which has been well established and continues to improve on the basis of different heating technologies. In addition, most of the studies proved that RT can be used to produce low-volume products. However, despite the potential advantages of RT in industry, there is a lack of studies and verification of the weld line and surface quality of plastic molded parts. Furthermore, the potential of RHCM in RT has not been explored, despite the use of RT in CIM having been investigated and research having been published. RT, on the other hand, is a great choice for low-volume production.

## 2. Injection Molding Process

The injection molding process is divided into several levels, i.e., operating cycles, including plasticization, filling, packing, and cooling [59,60,61,62]. The cooling stage is designated as the most significant of all stages due to its critical factors in developing productivity and molding quality, especially for complex molded parts [63,64,65]. Otherwise, defects such as sink marks, shrinkages, residual stresses, and warpage can occur if not addressed [66,67,68,69,70].

Initially, the plastic pellets are heated by screw shearing and a feed pipe before being plasticized to melt. The melt is then injected into the runner system to fill the mold cavities. After the cooling process, the molded part is ejected from the mold [38,71,72]. The quality of final molded parts is uncontrollable because it is influenced by many factors in the injection molding process. Some important factors which may have an impact on the finishing are the process parameters, mold design, and properties of plastic materials [52,73,74]. One of the factors that determines the quality of injection molded parts is the weld line [50,75,76]. The weld line refers to the process in which the liquid material is divided into two or more flows in the cavity during the injection process and merged together after a period of time [51,77,78,79].

There are five main phases in the injection molding process which are melting, filling, packing, cooling, and demolding [80,81,82]. It is desirable to keep the filling and packing phases at a high temperature to promote polymer melt flow and enhance reproducibility of the molded product [80,83,84]. A lower mold temperature is preferred during the subsequent cooling phase to allow for rapid cooling of the polymer melt, which occupies the majority of the molding cycle time [47,80,85,86]. As a result, to improve part quality and reduce molding cycle time, a dynamic mold temperature control system for rapid heating and cooling of the mold is required [80,83,87]. As shown in Figure 2, after the gate is solidified, the mold is cooled down rapidly by water to solidify and demold the polymer part [88]. However, in the CIM, the mold temperature control technique uses a concurrent cooling method, where a constant-temperature coolant circulates in a cooling channel to cool the mold and polymer melt [80,89,90]. Even though RHCM gives a longer cycle time, it improves the strength, eliminates the weld line, and improves the surface quality (high glossy surface) [38,91,92].

## 3. Rapid Tooling

Today, with different kinds of new technologies, global product development competition is becoming increasingly fierce. In addition, industries are constantly on the lookout for the latest technology that can compete to meet future demands, reduce costs, and produce small batches of high-quality products while also being able to meet sustainability goals [93]. In today’s market globalization, rapid tooling (RT) technologies have been steadily evolving with the demand for rapid product development in order to substitute conventional methods for better manufacturing processes, particularly for tool manufacturing [8,94,95].

RT is the AM technology that refers to the manufacturing methods of tooling [94,96,97]. RT utilizes the AM model as a pattern to construct a mold, or the mold for a small volume of prototypes is made directly from the AM process [8,27,98,99,100]. There are a variety of RT technologies accessible in the industry, such as the combination process of RT and AM, to lessen the time for making tools. Chua et al. [98] investigated several of RT and AM technologies using direct processes for tool production. Nevertheless, the majority of RT technologies implement indirect processes and models or master patterns produced by the AM process. According to Pouzada [101], there is an outline for the industry to take into consideration when choosing alternative production methods including RT, which includes a shorter delivery time, higher quality, shorter product development phase, and adaptation to global technology. This is a markedly new technique that has the potential to have a dramatic impact on the engineering practice, particularly during the product development stage, mostly in the casting and plastic molding industries [95,102]. As supported by Boparai et al. [27] and Mendible et al. [6], RT is also among the rapid prototyping (RP) variants, which have been proven to be cost-effective and time-efficient for the development of RT. 

According to Pontes et al. [103], RT differs from conventional tooling in that it takes much less time to fabricate tooling and can be classified into indirect or direct processes. Direct tooling techniques use AM processes to directly manufacture the molds without requiring a master pattern. The processes involved in direct tooling are SLS, DMLS, FDM, SLA, PolyJet, and 3D-P [8,96]. On the other hand, indirect processes utilize AM as master patterns to make a mold for castings. These processes include metal casting, resin casting, silicone rubber molding, and 3D KelTool process [6,97,100]. Jurkovic et al. [104] introduced the emergence of products and the evolution phase of RT technology that began in the early 1990s. Product design takes a long time. The standards start from the construction of the workshop, involving applications of computer-aided construction and geometric models, the rapid development of computer-oriented products (modeling and simulation) and virtual product development, rapid prototyping, rapid tooling, and the product development digitization process.

As the quantity of RT technologies with numerous techniques has increased, there has been a trend to categorize them, as can be seen in Figure 3. Indirect tools, direct tools, and casting patterns can be distributed into the implemented groups [98]. After all, the division of these groups may change depending on AM’s capability to speed up the production process. Some of the processes have been simplified with new technical capabilities, from indirect tooling to direct tooling, such as the use of mold inserts in an epoxy mold system rather than a conventional process [8]. In order to produce a good plastic injection mold, the selection of the material in mold making should be highly considered and understood.

## 4. Materials for Injection Mold

Several criteria need to be considered when selecting materials for the injection mold, including cycle time, production volume, cost, maintenance, materials to be molded, part requirements, manufacturability, and life cycle costs [69,105,106]. Hardness, capability to achieve the requisite surface finish, corrosion resistance, wear resistance, thermal stability, and thermal conductivity are all important properties of a material to consider when designing an injection mold [69,107].

Metals are the most frequently used material in the industry due to high machinability, great variety of composition and properties, heat treatment possibility, and high thermal conductivity [69]. Steel and aluminum are often used, contingent on the project’s requirements [69]. To ensure chemical resistance, strength, and hardness, steels are often made of chromium and nickel alloys, which make it a more cost-effective option [69]. On the other hand, aluminum alloy is a better thermal conductor than steel, reducing cycle time by up to 30% [69,107]. However, aluminum alloy molds only last around 2000 cycles, whereas steel molds last at least 50,000 cycles, making them sturdier than aluminum molds [69,107].

The mechanical properties of mold products do not change proportionally to the thermal conductivities or temperature of the mold material. The thermal conductivities of mold materials differ significantly, but changes in the mechanical properties of injected products from mold materials are barely noticeable. Steel mold material had four times the thermal conductivity of aluminum mold material in one study. When molding materials with different thermal conductivities were used, mechanical properties varied by 10–20% [108].

Usually, the mold is produced by a CNC machine with subtractive manufacturing. Harder materials including steel alloys necessitate specialized tools and further milling work, increasing the cost of the injection mold, as well as the cost of plastic parts to be produced [69].

### 4.1. Mold Base Material

A standard mold base is made up of four parts: the mold plate, the guide pin bushing, the return pin, and the screw. The mold plate is classified into two types according to its application: main plates (A plate and B plate) and structural plates (top clamping plate, bottom clamping plate, support plate, ejector plate, space block and runner stripper plate, etc.). Mold bases are made up of the A plate, B plate, and various structural plates assembled in a specific order [109].

The most common steel types used as mold base material are pre-hardened mold and holder steel, through hardening mold steel, and corrosion-resistant mold steel [110,111,112]. The pre-hardened mold and holder steels are mainly utilized for large molds, molds that do not require high wear resistance, extrusion dies, and high-strength holder plates [110,111,112]. These steels are usually supplied in a hardened and tempered state in the 270–400 HB range and do not require heat treatment [111]. Pre-hardened mold steel is usually utilized for large molds and medium production-run molds [111].

On the other hand, through hardened steels are mainly utilized for: long-term production, as well as specific molding materials with resistance to abrasion, high closing pressure or injection pressure, and suitability for high-pressure processes such as compression molding [111]. This type of steel is supplied in a soft annealed state. It normally undergoes rough machining, stress relieving, semifinished machining, hardening and tempering to the requisite hardness, finishing/grinding, and polishing or photoetching [111].

The choice of mold material may have an impact on the mold’s performance [111]. Different people define performance differently in terms of mold life, plastic part quality, and productivity. Wear, surface defects, deformation, and corrosion wear may occur as a result of reinforced plastic or long service life, and surface defects may occur due to polishing or EDM defects in the mold manufacturing process [111], while a plastic part’s quality is determined by its function, as well as its appearance. For highly polished molds, the choice of steel is significant. Steel should be clean and extremely low in impurities [111]. Due to the uneven temperature of the mold, tolerance may occur, which of course largely depends on the size and position of the cooling channel, as well as the mold material chosen. Mold materials with high thermal conductivity, such as aluminums or copper alloys, can be used in some cases. Sometimes, it is even possible to achieve productivity by selecting mold materials. The most obvious situation is to choose a material with high thermal conductivity [111].

The mold material requirements depend on the number of injections, the plastic material used, the mold size, and the desired surface finish; many different materials can be used. The following basic mold material properties must be considered: strength and hardness, toughness, wear resistance, cleanliness, corrosion resistance, and thermal conductivity [111].

A summary of the material properties commonly used as mold base materials is listed in Table 2. It can be seen that steel and aluminum are frequently used as mold base materials as they have high thermal conductivity and hardness, meeting the requirements of mold base materials in most studies [108,113,114].

### 4.2. Mold Insert Material

There are different types of mold inserts materials used in the fabrication of molds for the injection molding process. The fully hardened steel used for cavity and core inserts is typically held in a pre-hardened steel holder block (such as RoyAlloy or Ramax HH) [111]. Hardened molds or cavity inserts, for example, in the 48–60 Rockwell C range, can improve wear resistance, deformation and indentation resistance, and polishability [111]. When using filled or reinforced plastic materials, better wear resistance is particularly important. The ability to resist deformation and indentation in the cavity, gate area, and parting line contributes to maintaining the quality of the part. When a high surface finish on the molded part is required, better polishability is significant [111].

Aluminum is also a preferred metal for the fabrication of industrial mold inserts, although its strength and wear resistance [115] lower the benefits of aluminum. A good alternative for quick prototyping mold inserts is aluminum 6061-T6. It can be used very quickly [115] and subjected to diamond turning [116]. Its low specific density makes the molding machine easy to manufacture, assemble, and install, consequently reducing the production costs [115]. However, its strength is lower than that of copper-coated beryllium or nickel steels [115].

Steel alloys are often used for the production of molds because of their resilience, reliable mold operation, and long service lives, whereas their high strength impedes machinability and increases production costs [115]. Polishing of steel molds is often necessary [117,118]. Moreover, the rapid tool wear requiring the use of a special vibration-assisted cutting systems to reduce the cutting strength and wear in the tool [119] prevents steel alloys from being turned using a single point-diamond tool. Coating a nickel layer on steels can improve processability, but it also increases the cost and time of manufacturing [115]. The materials used for mold inserts in CIM, RHCM, and RT are discussed in further detail in the next section.

#### 4.2.1. Mold Inserts for Conventional Injection Molding

Mold making is an essential supplementary industry, and its related products correspond to more than 70% of consumer product components. The high demand for shorter design and manufacturing production lines, good dimensionality and product quality, and rapid changes in design has created bottlenecks in the mold industry [114]. It is a difficult process that necessitates the use of a skilled and experienced mold maker [114].

Mold maintenance requirements are reduced by ensuring that the core and cavity surfaces maintain their original finishing for long operation cycles. Classic stainless steel is the best choice for unacceptable production of rust and high hygiene requirements, such as in the medical industry, optical industry, and other industries requiring high-quality transparent parts [120]. Table 3 contains a summary of the physical and mechanical properties of mold insert materials commonly used in CIM. According to the summary in Table 3, steel is widely used as mold insert material in CIM due to its dimensional accuracy, surface finish, and productive capability required for plastic products [121].

#### 4.2.2. Mold Inserts for Rapid Heat Cycle Molding

Mold materials must be specifically chosen due to the rapid heating and cooling characteristics of RHCM. Compared with CIM mold materials, there are at least three additional factors to consider when selecting materials for electric heating molds. First, as the temperature of a low-heat-capacity material changes easily, a mold material with this characteristic is preferable for electrically heated RHCM. Secondly, because the mold requires frequent heating and cooling, fatigue cracks caused by cyclic thermal stress are more likely to appear on the surface of the cavity compared to other molding processes. Therefore, mold materials with a low thermal expansion coefficient and high thermal fatigue strength are needed.

Furthermore, due to the high mold temperature, several corrosive gases such as hydrogen chloride (HCl), hydrogen fluoride (HF), and sulfur dioxide (SO_2_) generated by the melt decomposition will corrode the surface of the cavity. Thus, the cavity/core material must be highly corrosion-resistant. High-thermal-conductivity materials, such as copper, copper alloys, and aluminum alloys, are often used as cooling plates for electric heating molds to enhance the cooling efficiency [127].

Steel is often chosen as the mold material. Steel has a lower heat transfer than aluminum plates (due to its higher density and lower conductivity), although steel is more commonly used in large series (often up to 106 cycles) because aluminum has poorer mechanical properties under injection pressure (not only strength but also stiffness) [128].

Table 4 summarizes the physical and mechanical properties of the mold insert used in the RHCM. A higher thermal conductivity of the cavity/core material results in higher heating and cooling efficiencies and a shorter molding cycle time [87].

#### 4.2.3. Mold Inserts for Rapid Tooling

Rapid tooling is among the rapid prototyping applications in the manufacturing industry. It allows building molds for small batch production products quickly and at low cost. It can be divided into direct and indirect tooling, as well as hard and soft tooling. Direct tooling is attained directly through a rapid prototyping process which uses soft materials (such as stereolithography materials) [3,9,15,129,130,131]. Several other tools are made of hard materials, such as powder metal [9,132], while, in the indirect tooling method, a casting pattern is produced through a rapid prototyping process and then used to build the necessary tool. Aluminum-filled epoxy resin [103,133] is a prevalent soft material because of its ease of use in fabricating mold inserts. Silicon rubber [134] is mostly used to make indirect tools.

The tool life is the most important consideration for injection molds formed using the rapid tooling approach. Since rapid prototyping technologies have matured, tools directly produced by rapid prototyping machines can accurately and precisely depict all the specific details and features of the model. Nevertheless, certain soft rapid prototyping materials have lower thermal conductivity and are typically unable to withstand high injection pressure and melt temperature [134]. Therefore, the lifespan of the tool is shorter. Even though additional processes such as metal laser sintering can be used to coat soft materials with a layer of metal [135] to increase its hardness, they increase the difficulty of the manufacturing process. In contrast, epoxy resin is a commonly used material in indirect tooling method because it can be easily molded with the casting pattern. The addition of metal powder can significantly increase its hardness and thermal conductivity, thereby further extending its tool life. Nevertheless, it also makes the epoxy resin mold cavity brittle. Thus, tools created through the indirect tooling method do not last long [9]. Some of the studies focusing on rapid tooling are presented in Table 5.

Tomori et al. [136] investigated the impact of mold performance and part quality on the composite tooling board material formulation and processing parameters. A sample preparedness flow chart is displayed in Figure 4. The boards were made of three ingredients: RP4037 (fluid), RP4037 hardener, and silicon carbide (SiC) filler (powder). Three levels of tooling board formulations were chosen for the six molds: 28.5, 34.7 and 39.9 wt.% of the SiC filler, along with two cutting speed levels (1.00 and 1.66 m/s). The variable parameter of this study was cutting speed, whereas the surface roughness of molded parts was the response variable. Since no apparent mold damage was found, the physical structure of the mold was unaffected by SiC concentration and cutting speed. This finding concluded that SiC concentration in the mold greatly impacts the surface roughness of the molded parts. In addition, the flexural strength increased (58.75 to 66.49 MPa) as SiC filler concentration increased, following a pattern similar to the mold material’s thermal conductivity. However, this investigation did not discuss the consequence of filler concentration on the weld line of molded parts.

Senthilkumar et al. [137] studied the mechanical behavior of aluminum (Al) particles added with epoxy resin. A specimen was cast using various percentages of Al filler mixed into the epoxy resin. Optical microscopy results showed that the distribution of Al particles in the epoxy resin matrix was uniform. According to this finding, increasing the proportion of Al particles in epoxy resin matrix resulted in a significant increase in thermal conductivity (3.97 to 5.39 W/m·K) and hardness value (69 to 89 R_HL_). However, fatigue life of the specimen decreased from 15,786 to 734 cycles as the percentage of Al in the epoxy resin increased. The best proportion to improve mold durability and performance was 45–55 wt.% Al filler particle. This parameter improved in terms of hardness, fatigue life, and thermal conductivity by 72 R_HL_, 10,011 cycles, and 4.06 W/m·K, respectively. The fatigue life could be reduced by 36.58% for every 5 wt.% increase in Al filler particles, while the hardness value increased by 4.34%. Nonetheless, no further research has been conducted on flexural strength, compressive strength, tensile strength, and surface appearance of the produced molded parts.

Srivastava and Verma [34] carried out a study on the impact of particles on the mechanical properties of reinforced epoxy resin composites containing Copper (Cu) and Al particles. Cu and Al particles were mixed separately in epoxy resin as fillers in different compositions (1, 5, 8, and 10 wt.%). The mechanical test results showed that Al-reinforced epoxy resin has great tensile properties, i.e., 104.5 MPa at 1 wt.%, while Cu filler epoxy resin composites performed the best in the hardness test (22.4 kgF/mm^2^ at 8 wt.%), with a compressive strength of 65 MPa at 10 wt.%. In addition, Cu-filled epoxy resin composites had a lower hardness than Al-filled composites, but still performed better than Al. This finding concluded that tensile strength and wear loss decreased steadily as filler content increased, whereas hardness, compressive strength, and friction coefficient increased as Cu and Al filler weight percentages increased. In short, the effect on the weld line on the surface of the molded parts was still lacking in this study.

Fernandes et al. [35] investigated the mechanical and dimensional features of molded PP injection parts for rapid tooling in epoxy resin/Al inserts. The circular geometrical component used for the work had a diameter of 140 mm and five central cavities connected by a 2 mm diameter segment. The runner was 60 mm long, the entrance diameter was 6.5 mm, and the draft angle is 2°. In this study, a new hybrid mold made of epoxy resin and Al was used to inject polypropylene (PP) parts to test the proposed mold. Moreover, similar parts were injected using an AISI P20 (ordinary mold) steel mold, just like the real application. While the tensile strength at yield of the parts injected with epoxy resin/Al inserts (22.0 ± 5.0 MPa) was higher than that of the parts injected with injection steel AISI P20 inserts (20.0 ± 4.5 MPa), this difference was not statistically significant. Other properties (ultimate tensile strength, elongation at break, and modulus of elasticity) in epoxy resin/Al-injected parts were lower than in steel AISI P20-injected parts. Furthermore, the Shore D hardness of parts molded with AISI P20 steel inserts increased by 8.5% compared to epoxy/Al inserts. The geometric deviation of parts injected with AISI P20 steel mold showed less shrinkage than parts injected using the epoxy/Al mold. According to these findings, epoxy/Al molding blocks could be a high-quality substitute for rapid tooling in the production of small series of products. Moreover, the findings with regard to the effect on the weld line of molded parts produced was not discussed in this investigation.

Khushairi et al. [138] analyzed the mechanical and thermal properties of various Al-filled epoxy compositions with the addition of Cu and brass fillers. Different compositions of brass and Cu filler were mixed in Al-filled epoxy (10, 20, and 30 wt.%). At the highest filler composition, the density of brass and Cu was 2.22 g/cm^3^ and 2.08 g/cm^3^, respectively. Adding 30% Cu fillers to an epoxy matrix resulted in the highest average thermal diffusivity (1.12 mm^2^/s) and thermal conductivity (1.87 W/m·K), whereas adding brass had no effect on thermal properties. The compressive strength increased from 76.8 MPa to 93.2 MPa when 20% brass filler was used, and from 76.8 MPa to 80.8 MPa when 10% Cu filler was used. Due to the presence of porosity, further addition of metal fillers reduced the compressive strength. This finding concluded that fillers improve the mechanical, thermal, and density properties of Al-filled epoxy. Nonetheless, an extensive study on the surface appearance, particularly the weld line of the molded parts, is needed to analyze the quality of the molded part.

Kuo and Lin [139] investigated the manufacture of Fresnel lenses using rapid injection molding with liquid silicone rubber. The experiment was developed with RT and LSR parts for the development of a horizontal LSR molding equipment (Allrounder 370S 700–290, ARBURG). Al-filled epoxy resin could be used to produce injection molds for LSR injection molding. The Al-filled epoxy resin mold’s average microgroove depth and width had replication rates of 90.5% and 98.9%, respectively. The average microgroove depth and width transcription rates of LSR molded parts were approximately 91.5% and 99.2%, respectively. The microgroove depth and width variations of LSR molded parts could be controlled within ±1 µm. After 200 LSR injector test runs, the average surface roughness of the Al-filled epoxy resins improved by approximately 12.5 nm. However, further tests on the tensile strength, compressive strength, hardness, and density, as well as observations of the weld line, are required to understand the impact of rapid injection molding on the proposed mold in terms of the quality of the molded parts.

According to the above review, several aspects (flexural strength, hardness, thermal conductivity, tensile strength, compressive strength, density, thermal diffusivity, and surface roughness) are important in manufacturing new potential mold inserts material for the injection molding process, as well as in the rapid tooling technique. However, most previous studies did not consider an observation of the surface appearance (weld line) of the molded parts when attempting to find the best proportion of materials in manufacturing mold inserts. Findings from previous research have shown that, by implementing a suitable material, mold inserts can achieve excellent machinability and high compressive strength, combined with sufficient toughness, good resistance to heat and wear, and high thermal conductivity. Nevertheless, the implementation of the rapid tooling technique in rapid heat cycle molding (RHCM) is yet to be investigated in terms of its contribution in the molded parts produced.

## 5. Rapid Heat Cycle Molding

In conventional injection molding, the mold temperature is significantly lower than the glass transition temperature or melt temperature of the polymer to reduce the molding cycle and enhance output efficiencies. Under such molding conditions, it is inevitable that the polymer melt will solidify prematurely during the filling stage, resulting in various molding defects such as short shots, flow marks, weld marks, jetting marks, swirl marks, and fiber-rich surfaces. In RHCM, rapid mold heating and cooling technology must be implemented to heat the cavity surface in a reasonably wide temperature range. Before performing melt filling in RHCM, the cavity surface is rapidly heated to a much higher temperature than CIM, typically above the glass transition temperature or polymer melt temperature. Therefore, the phenomenon of premature freezing of the polymer melt in the filling stage can be eliminated in the RHCM, thereby eliminating the frozen layer formed in the CIM. As a result, RHCM will effectively resolve the aforementioned inherent CIM molding defects. To shorten the molding cycle, the injection mold and shaped polymer melt cools faster in RHCM than in CIM after the filling stage [37].

Before the filling process in RHCM, the mold is rapidly heated to above the glass transition temperature, and then molten plastic is injected into the cavity. The mold is rapidly cooled after the filling stage to harden the molten plastic. After the cooling stage, the plastic parts are finally ejected [76]. The solid line in Figure 5 shows a demonstrative example of a mold temperature profile, where Tg denotes the material’s glass transition temperature. The heating time is indicated by the symbol t_h_. The injection, packaging, and cooling times are denoted by the letters *t_inj_*, *t_p_*, and *t_c_*, respectively [76].

## 6. RHCM versus Conventional Injection Molding

The rapid heating cycle molding (RHCM) injection method has been used to overcome product surface defects generated through conventional injection methods, such as weld marks and flow marks, by contrasting the advanced technology with conventional injection methods and products formed [56]. In this section, three comparisons are made in terms of the working process, mold design manufacture, and part quality.

The polymer melt is thoroughly combined during the injection molding process because RHCM uses high temperature, along with a rapid heating and cooling method. The melt flows much better than in the conventional injection method, with lower viscosity. The surface of the mold cavity is made with a high-gloss surface. As a result, the product surfaces are bright and smooth as a mirror, with no weld marks or flow marks.

In general, the conventional injection molding process consists of five stages, namely, the plasticization stage of the polymer, the injection stage, the filling stage, the cooling stage, and the ejection stage. Once the polymer has been plasticized into a molten state, it is injected at a higher pressure and speed into the mold cavity through a nozzle. Following the packing stage, the polymer melt is cooled to a low temperature using water. After that, the mold is opened, and an ejector pin is used to eject the polymer. At this point, the conventional injection molding process has completed one cycle before beginning the next cycle [56].

In contrast to CIM, RHCM injection technologies can be classified into six phases. The stationary mold is heated prior to injecting the polymer melt into the mold cavity. During this stage, the surface of the mold cavity is heated to the injected polymer’s glass transition temperature. The cavity is then filled with the polymer melt. The cavity temperature remains unchanged during the subsequent packing process until the cooling phase starts. Cooling water is utilized as a coolant to preserve the mold and polymer melt inside it to a certain low temperature. The plastic part can now be ejected to complete the RHCM injection cycle. The cavity surface is heated quickly again before the next injection process begins. Then, the next RHCM cycle of injection starts [56].

In RHCM, the initial mold cavity temperature is significantly higher than that in conventional molding, and the temperature is significantly different. However, since the mold temperature rises and falls in a short period of time, the molding cycle is nearly equal to that of a conventional injection molding process [55]. To shorten the molding cycle time, the heating stage for the RHCM mold, which requires heating of both the cavity and the core sides, can begin at the same time as part ejection. If only the cavity side of the mold needs to be heated, the mold can be heated as it opens. During the heating stage, the mold is heated to a preset high temperature, which is usually higher than the polymer’s glass transition temperature [87]. Figure 6 shows the RHCM technology principle and the RHCM mold’s temperature variation and injection cycles for conventional injection molding. In this figure, a is the heating stage, b is the injection and packing stage, c is the cooling stage, d is the mold opening and part ejection stage, and a’, b’, and c’ are the next stages of the injection cycle, while line 1 represents the heat distortion temperature of polymer, line 2 is the mold temperature of RHCM, line 3 is the ejection temperature of the part, and line 4 is the mold temperature in the conventional injection method.

This figure shows the mold temperature changes during RHCM injection cycles, including the polymer’s heat distortion temperature and the ejection temperature of part. Although conventional molds are still at the same temperature, the RHCM technology work process is more complex than the conventional injection process. Furthermore, the temperature control of RHCM molds is much more complicated than that in conventional molding, and it is normally controlled by conventional temperature controllers [56].

### 6.1. Application of RHCM in CIM

Some defects in plastic parts produced by conventional injection molding (CIM) can be solved by RHCM, such as flow marks, silver marks, jetting marks, weld marks, exposed fibers, and short shots [43]. However, RHCM is not a solution for all injection-molded part flaws. Warpage is one of the defects that cannot be solved using RHCM. Li et al. [88] investigated a method for predicting the warpage of crystalline parts molded using the RHCM process. To predict warpage, multilayer models with the same thicknesses as the skin–core structures in the molded parts were developed. The thicknesses of the layers varied with each molding process. The upper skin layer of each injection-molded part was heated on the stationary side. The prediction results were compared to the experimental results, which revealed that the average errors between predicted warpage and average experimental warpage were respectively 7.0%, 3.5%, and 4.4% in CIM, RHCM under a 60 °C heating mold (RHCM60), and RHCM under a 90 °C heating mold (RHCM90). Apart from that, the microstructure and temperature in CIM were symmetrical along the thickness direction and asymmetrical in RHCM. The predicted warpage was influenced by crystallinity, and the warpage predicted with crystallinity was greater than the warpage predicted without crystallinity, especially in RHCM-molded parts.

Furthermore, Li et al. [48] investigated the effect of the weld line on the tensile strength of RHCM and CIM components. Tensile testing results showed that, for specimens with a weld line, the strength of the RHCM specimen was significantly higher than that of the CIM specimen due to the smaller dimensions and higher bonding strength of the weld line. Moreover, the RHCM specimen’s weld line was smaller than that of the CIM specimen. Furthermore, the two melt fronts that formed the weld line were welded together better due to the higher temperature and pressure in the RHCM process compared to the CIM process, resulting in a higher bonding strength of the weld line. The tensile strength of specimens without a weld line showed no discernible difference between the two molding processes. In addition, a thin surface layer of material containing the V-notch at the weld line was removed from the RHCM and CIM parts to investigate its effect on tensile properties. When compared to specimens with the original full thickness, thinned out specimens with a weld line showed improved tensile strength, while specimens without a weld line showed decreased tensile strength.

On the basis of the results from the previous researchers [48,88], it can be said that the application of RHCM in CIM has a big impact on the molded part. Previous studies used the same mold insert for both molding processes. The part molded by RHCM showed greater warpage predicted with crystallinity compared to that predicted without crystallinity. Additionally, the strength of the RHCM molded part was higher than that of the CIM molded part with the same shape and size, while the weld line of the RHCM molded part was smaller compared to that of the CIM molded part. Thus, it can be concluded that using RHCM in CIM yields a significant difference in the molded parts produced, demonstrating that RHCM produces a better result.

### 6.2. RHCM Technologies

Many studies have been performed to enhance the surface quality of plastic products by optimizing the process parameters. Contrary to popular belief, optimized parameters do not eliminate defects, but improve the surface quality of the molded parts. It was recently discovered that raising the mold surface temperature eliminates defects, increases flow length, and improves surface quality [43,87,140,141,142,143,144]. RHCM is an innovative technology that allows for dynamic mold temperature control during the injection molding process. RHCM technology requirements for the temperature of the heat distortion for the injecting polymer should be met before the mold cavity is injected. At the ejected temperature, it must be quickly cooled down. The difference in the mold temperature is significant. Therefore, if the cooling and heating methods are the same as in conventional methods, the production cycle must be extended. The RHCM mold is heated and cooled rapidly to ensure heating and cooling efficiency. Consequently, the structure of the mold differs from a conventional mold, as does the heating method. Companies such as Foreshot Industrial Corporation, Taoyuan City, Taiwan and Letoplast S.R.O., Letovice, Czech Republic have developed and applied RHCM technology, offering the benefits of a perfect product with a glossy surface that does not require painting [145,146]. Letoplast specializes in the production of visual and technical plastic parts made of PC-ABS, ABS, PP, PPE, PC, PA6, and other materials [146]. RHCM technology is used in plastic parts such as network communication equipment computers/communications/consumer electronics, appearance parts, and liquid crystal display televisions (LCD TVs) [145]. This technology allows engineers, technologists, or mold designers to avoid unnecessary premature melt freezing throughout the filling stage, which lowers the melt flow resistance and improves the molten plastic filling capability [38].

Chang and Hwang [147] proposed an infrared heating method for the mold cavity surface. For thermal surface condition assessment, a transient thermal simulation was developed. A change in the mold structure is not necessary with this method. This method has a high construction cost, and it is hard to consistently heat the longitudinal or complex mold cavity. Fischer et al. [148] investigated the effect of a locally different cooling behavior on the injection molding process of semicrystalline thermoplastics. An innovative dynamically tempered mold technology with different temperature zones was investigated, allowing the production of thin-wall components with locally different component properties. The preliminary results showed that, by influencing the inner component properties, significant differences in the optical and mechanical component properties could be achieved. On the other hand, Yao and Kim [142] evaluated the benefits of heating the mold surface through the use of a thin metal coating and thermal insulation. Copper poles were used to control the temperature by heating the thin metal layer. This method was shown to be efficient in terms of energy usage and temperature control. However, the complex coating and tiny parts of the cavity make this process more challenging; in addition, there are safety hazards, such as in the resistor layer’s insulation, with the resistor having a limited life time. Table 6 shows some of the studies conducted on RHCM technology.

Chen et al. [39] studied the feasibility, efficiency, and effects of induction heating on the weld line surface appearance for dynamic mold surface temperature control. A simulation and experiment were used to analyze the induction heating with the spiral coil coinciding with the coolant cooling to evaluate its rapid heating/cooling ability and the uniformity of the mold plate surface temperature. Initially, ANSYS 3D thermal analysis was performed on a mold plate with a heating source to verify the analytical capabilities and accuracy. Then, induction heating was performed on the double-gate tensile sample mold to control the surface temperature of the acrylonitrile butadiene styrene (ABS) melt injection mold and determine its influence on the surface marking and weld line strength. From the results, it was indicated that it took 3 s for the center temperature of the plate to rise from 110 °C to 180 °C and 21 s to rise to 110 °C during induction heating and cooling, while, for the second heating/cooling cycle, it took 4 s to reach 200 °C and 21 s to return to 110 °C. The surface temperature of the mold plate could be elevated to around 22.5 °C and cooled to 4.3 °C. The study also proved that the temperature-controlled induction heating of the mold surface also contributed to the elimination of surface markings and to enhancing the strength of the weld line on injection molded ABS tensile bars. However, this study did not yield a positive correlation between the heating time by induction heating and the temperature distribution pattern on mold plates produced by the rapid tooling technique. Therefore, further investigation is required.

Huang and Tai [43] studied the use of induction heating to rapidly heat the mold surface during the injection molding process, thereby improving the microstructure replication effect of light-guided plates (LGP). First, oil was used to heat the mold to 80 °C, and then induction heating was used to heat the surface of the LGP mold above the glass transition temperature. Induction heating was implemented to achieve the high mold temperature essential for the short filling stage throughout the microstructure replication process of LGP, as well as to rapidly lower the mold temperature in the cooling process, allowing the molded part to cool faster. The heating coil was a single copper wire used for small area heating. The geometry of the induction coil is presented in Figure 7. The molded part of PMMA plastic was a flat LGP of 40 mm × 30 mm, with length and width of 1 mm. Results from these studies showed that the induction heating system only achieved a temperature uniformity of 6.7–12.7 °C. In addition, the combination of an oil heater (80 °C) and an induction heating system (up to 110 °C) offered the best replication performance, significantly raising the replication ratio up to 95%, which was 6% higher than when the mold was heated with an oil heater and 7.8% higher than when the mold was left unheated. Thus, these studies prove that induction heating can improve the LGP microstructure filling efficiency. However, in this study, only the performance of the heating technology when heating the mold surface was discussed, without further details on the strength and surface appearance (weld line) of the molded parts produced. Therefore, a further investigation of the correlation between heating technology and the properties of the molded part produced is required.

Huang et al. [44] used the same experimental setup as Huang and Tai [43] to perform rapid induction heating on the surface of a 2 inch LGP mold, but with a different coil design, using three-layered copper wire, to study the influence of high mold temperature in induction heating, as shown in Figure 8, to enhance the replication rate of LGP microfeatures. The temperature profiles of the Ni heating plate used as the mold insert in 2 inch LGP were investigated in order to identify the optimal power rate setting in the induction heating system; these temperature profiles correlated with the different power rates. Additional testing was carried out to determine the efficiency of the optimal process parameters defined by the Taguchi method. The ANOVA results indicated that the most significant parameters affecting the replication ability of microfeatures were the mold surface temperature and injection speed. As the surface temperature of the mold increased to 150 °C, the replication rate of the average nine-point height of the microfeatures and the average nine-point angle increased to 91.6% and 98.1%, respectively. However, the effects on the strength of the molded parts produced and the temperature distribution surface of the mold insert were not discussed in detail in this study.

Tsai et al. [46] reported the vapor chamber’s effect on part tensile strength, which was made using plastic (ABS) material in an injection molding process. This experiment was developed with a copper heat pipe of the flat-plate type. Water was used as the working fluid. Then, to circulate the working fluid, a copper mesh formed using 50 μm wire with 100/200 mesh pore and a solid copper cylinder with a diameter of 2 mm were used to support the vacuum and loading forces. The experiment used the shape of the specimen shown in Figure 9 for tensile testing. The experiment was conducted with two distinct mold designs, one with a single gate and another with two opposite gates. The cavity temperature was used as a parameter in the experiment, while the output was the molded part’s tensile strength. As a result of the weld line, the tensile strength of the test part formed by the two gate/no vapor chamber heating system was 11.1% lower than that of the test part formed by the one gate/no vapor chamber heating system (Figure 10). The two gate/vapor chamber system produced a fine weld line, as well as a tensile strength 6.8% higher than that of the two gate/no vapor chamber system. Furthermore, the tensile strength increased by 3.2% when the preheating temperature increased from 75 to 110 °C in the two gate/vapor chamber system. Due to the extended fluid flow, the use of the vapor chamber heating system and the increase in the preheating temperature seemingly improved the tensile strength of parts molded with two opposing gates. However, despite the fact that this study focused on conventional mold steel, additional research on the effect of heating on the mold insert produced by the rapid tooling technique should be considered.

The heating and cooling system designs of electric heating molds are critical for RHCM with electric heating because they directly affect the heating/cooling efficiency and temperature uniformity, which have a decisive influence on the molding cycle and part quality [127]. Electric heating in RHCM can be accomplished through the deployment of electrical heating rods and cooling tunnels in the stationary mold, which can be rapidly heated via electric heating prior to filling and cooled via circulating water after filling [50,88]. In another study, Wang et al. [54] investigated the thermal response of the electric heating with the deployment of a cartridge heater in rapid heat cycle molding, as well as its impact on the surface appearance and tensile strength of the molded part. ABS/PPMA and fiber-reinforced PP were the materials used in the molded part experiment. The mold material used for the mold holder was HT 350, while that for the mold plate was AISI 1045, and that for the cavity plate was AISI H13. ANSYS thermal modules were used to investigate the temperature distribution and thermal response of the cavity surface. The thermal cycling process caused the surface temperature of the cavity to rise gradually. The maximum cavity surface temperature in thermal cycling could reach 185 °C after 30 experiment iterations with 60 s of mold heating time and 20 s of mold cooling time. This implied that the power density of the heater must be high enough to shorten the RHCM molding cycle by reducing the mold heating time. As the power density of the heater increased from 15 to 30 W/cm^2^, the heating time of the cavity surface to be heated from 30 to 120 °C could be decreased from 58 to 19 s. Experimental results showed that, compared with CIM, the RHCM method reduced the tensile strength of parts without the weld line (Figure 11). In the case of ABS/PMMA plastics, the tensile strength of the weld line part was also significantly lowered due to RHCM. The RHCM method enhanced the weld line factor for ABS/PMMA and 20% glass fiber-reinforced PP. Findings were limited to the effect of RHCM heating technology on the cavity surface and the parts molded using conventional mold inserts. Nonetheless, more research into the effect of RHCM heating technology on mold inserts produced by rapid tooling technique is needed.

Wang et al. [47] studied the design of heating/cooling channels for an automotive interior part, as well as its evaluation in rapid heat cycle molding. Polycarbonate (PC) material was used for the molded part. The general commercial ANSYS was used to simulate the transient thermal response of the mold cavity throughout the heating process to analyze the efficiencies of the original heating/cooling channel design. The designed heating/cooling systems with baffles could significantly improve the cavity surface thermal response efficiency. Steam was used at a temperature of 180 °C and a pressure of 1.0 Mpa. When experimental and simulation results were compared, the reasonable heat transfer coefficient was determined to be 14,000 W/m^2^·°C. Heat exchange occurred between the cavity plate and the environment via natural air convection. The convective heat transfer coefficient was determined to be 25 W/m^2^·°C. According to the results, the mold time constant could be lowered from 5 s to 2.5 s, resulting in a 50% decrease in time. Heating efficiency could be improved by 27.3%. The total surface temperature differences could be reduced to 20–30 °C. Moreover, by increasing the temperature of the cavity surface just prior to filling, the weld marks on the surface of the part could be significantly reduced (Figure 12). The weld marks could be completely removed as the cavity surface temperature approached a critical level. The critical cavity surface temperature of the plastic material used was approximately 130 °C. Findings from this study proved that the surface gloss of RHCM products was more than 90%. However, further studies on the implementation of RT in RHCM are required.

Nian et al. [45] studied the key parameters and optimized design of single-layer induction coils for external rapid mold surface heating by controlling the process parameters of induction heating (suitable for applications involving mold plates with different thicknesses and coil positions). Figure 13 depicts the spatial dimensions used in the simulation. An experiment was performed to validate the thickness of the heated workpiece (SKD61) and the induction coil (copper) position influencing the heating rate and temperature uniformity. Furthermore, the Taguchi method and principal component analysis were used to identify the best control factor combination to obtain high heating rates with low temperature deviations. According to the simulation results, the position of the induction coil and the thickness of the workpiece were indeed important design parameters. The optimal parameters indicated heating a 10 mm thick workpiece when designing a single-layer induction coil with a 15 mm turn pitch at a distance of 5 mm from the heating target. Additionally, the coil position should not be offset, and the operating frequency and waiting time must be set to 35 kHz and 6 s, correspondingly. Furthermore, the results of the experiments showed that an optimized design of the induction coil outperformed conventional single-layer coils in terms of temperature uniformity. The heating rate increased from 3.3 °C/s to 4 °C/s (an increase of 19.5%), and the heating uniformity increased by 62.9%. In terms of model validity, all MAPE values indicated high prediction accuracy. However, other important tests such as strength tests, as well as an observation of the appearance of the molded part produced by the induction heated mold surface were not performed.

Li et al. [48] investigated the effects of the weld line and its structure on tensile strength, as well as its process dependence, in CIM and RHCM. The specimens were made of isotactic polypropylene (iPP). Figure 14 shows the dimensions of the dumbbell-shaped specimen; the thickness was 2.5 mm. An RHCM mold with a maximum clamping force of 3800 kN, a maximum melt volume of 1239 cm^3^, and a maximum injection pressure of 182 Mpa was used to mold the CIM and RHCM parts. Electrical heating rods were used to quickly heat the mold before filling it; after filling, cooling tunnels cooled the mold with circulating water. The only parameter used for the experiment was the mold temperature, and the output was the tensile strength of the weld line on the molded part. According to the results, the RHCM process reduced the dimensions of the weld line, particularly the width of the weld line adjacent to the heated stationary half. The tensile strength of dumbbell-shaped specimens cut from RHCM parts with and without a weld line is shown in Figure 15. Due to the presence of skin layers or weld lines on both RHCM and CIM specimens, their trends were similar. The RHCM specimen with the weld line had a higher tensile strength than the CIM specimen, but the strength without the weld line was remarkably similar. The weld line reduced the overall tensile strength, while RHCM reduced the decrease in tensile strength. Meanwhile, in the RHCM process, there was no discernible influence on the tensile strength of the specimens not having a weld line. The removal of the surface layer improved the tensile strength of specimens with weld lines but reduced it in specimens without a weld line. Findings from this study prove that RHCM can reduce the effects of weld line on the tensile strength. Nonetheless, further studies on the potential of RT in RHCM are yet to be conducted.

Xie et al. [49] studied the thermal response of the graphene layer, as well as the impact of rapid thermal cooling with carbide bonded-graphene coating on molded product defects such as weld lines, internal stress, and replication fidelity. In this study, the heaters and heating channels implanted in the CIM were detached. They could be rapidly heated to a critical temperature by fine-tuning the voltage applied to the carbide-bonded graphene coating layer, or they could be cooled by a coolant channel. The semicrystalline polymer PP was used for the weld mark experiments, while PS was used in both the residual stress and the replication fidelity experiments. According to the results, with increasing voltage, the rate of heating of the insert surface increased. Once the voltage was set to 240 V, the coating heated up to 145.6 °C in 10 s, indicating that the average and transient heating rates could reach 11.6 °C/s and 16.1 °C/s, respectively. Therefore, the graphene coating could act as a thin-film cavity resistance heater, heating the polymer above Tg in seconds. Additionally, experimental results indicated that improving the surface temperature of the silicon insert could reduce the width of weld lines on molded products. If the temperature was high enough, these lines could also disappear entirely. Figure 16 shows that tensile strength and tensile elongation yields increased with silicon insert surface temperature. The average tensile strength and tensile elongation yields for silicon inserts at room temperature were 27.14 MPa and 128.30% respectively. However, at 140 °C, the average tensile strength and tensile elongation yields increased to 37.39 MPa and 468.77%, correspondingly. Findings from this study concluded that the rapid thermal cooling method could produce sample plates with uniform size thickness (600 µm) and mold products with fewer weld marks, less minimal internal stress, and enhanced replication fidelity. The tensile strength and tensile elongation yield of the products were increased by 37.77% and 265.11%, respectively, with even less energy consumption. Nonetheless, more research on the temperature distribution of the cavity surface of the mold insert is needed.

Liu et al. [50] studied the microstructure, tensile properties, and surface quality of glass fiber-reinforced PP parts molded by RHCM, using PP with 30% short glass fibers. The employed electric heating RHCM apparatus consisted of electric heating rods, a K-type thermocouple sensor, and a mold temperature controller MTS-32II system for measuring, regulating, and displaying the mold surface heating temperature. In the sample thickness direction, Autodesk Moldflow was used to evaluate the glass fiber orientation distribution. The simulation’s material properties and boundary conditions were based on the CIM and RHCM experiment parameters. RHCM60, RHCM90, and RHCM120 were the sample mold part designations for the cavities heated to 60, 90, and 120 °C. The tensile tests were conducted according to ASTM D638, and the results showed that, as the cavity surface temperature increased prior to filling, the tensile strength increased first, before decreasing. Tensile strength was the highest in the RHCM60 sample, followed by the RHCM90 and CIM samples, while it was lowest in the RHCM120 sample (Figure 17). These variations were due to the microstructure of GFRPP composites being highly dependent on the cavity’s surface temperature. As the cavity surface temperature increased, the thickness of the skin layer with medium fiber orientation along the flow direction decreased. RHCM samples had a thicker shear layer than CIM samples, with RHCM60 having the thickest shear layer. Therefore, the overall crystallinity and fiber orientation of RHCM60 in the cross-section were greater than those of the CIM sample. When an external force applied tensile load to the sample, the increase in fiber orientation caused more fibers to bear the load transmitted by the substrate. The increase in matrix crystallinity indicated an increase in the contact area between the crystal and amorphous regions, which aided in load transfer from the matrix to the fiber, resulting in RHCM60 having a higher tensile strength than the CIM sample. With a weak fiber orientation, the core layer thickness decreased initially, before increasing; the RHCM120 and RHCM60 samples had the thickest and thinnest core layers, respectively. It was also reported that the sample’s surface gloss tended to increase as the surface temperature of the mold cavity increased, but the changes in surface gloss were noticeably reduced as the mold heating temperature increased above 90 °C. Furthermore, this study took into account the temperature distribution on the mold cavity surface of various mold types of mold inserts in CIM and RHCM. 

According to the above review, RHCM has the potential to eliminate weld lines and improve surface gloss. The main distinction between RHCM and CIM is that the mold is heated and rapidly cooled to the ejection temperature after the filling process. The mold is heated above the molten plastic’s glass transition temperature to achieve the optimum surface quality on the molded parts. In the reviewed studies, electric heating, induction heating and steam heating were extensively used as heating methods in RHCM in order to heat the mold in the quickest time. According to the research on parts produced by RHCM, it was found that defects caused by weld lines not only affect the surface appearance, but also affect the strength of the molded parts. A lower strength is obtained due to the stress concentration point between the V-notch and the weak bonding area, and the bonding interface strength at the weld line is not as strong as that in the matrix. In addition, different levels of mold surface temperature were introduced into RHCM research to obtain the optimum molded part quality. However, there is a lack of studies on the temperature distribution of the surface of the mold insert, which will give further information on the effect of heating on the parts produced. Nevertheless, most of the research only focused on using conventional materials as mold inserts, and there is still lack of support for and verification of the application of RT in producing mold inserts that are less expensive than conventional material molds in the RHCM method. The next section in this paper focuses on the effects of RCHM on molded parts. As confirmed in previous research, the RHCM method can increase the strength of the molded parts by reducing weld lines.

## 7. Effect of RHCM on the Molded Parts

Weld lines reduce the strength of injection molded parts; therefore, removing them is important for producing materials with appropriate structural integrity. However, this has not been implemented so far. Consequently, some researchers conducted studies to reduce the four adverse effects of weld lines. The following methods were proposed: optimizing the composition of the materials, improving the design of the molds, and changing the parameters of the injection molding process [39,150]. Furthermore, it is important to detect weld line defects because they may occur in unexpected areas. During the injection molding process, two or many flows meet, while the thickness and temperature may change. Destructive testing can identify these weld line flaws. If no obstacles exist, such as unforeseen circumstances in molding conditions (flux and temperature) or uneven materials during injection molding, for a large number of identical products, weld line defects are expected to occur at the same position [151]. Thermal assistance has become a popular technique for improving the mechanical properties and aesthetic properties of molded products during the injection molding process. RHCM is the most famous thermal assistance method to improve the weld line strength and surface defects (Table 7).

Wang et al. [37] investigated the mechanical and aesthetic properties of molded parts by investigating the effects of cavity surface temperature during the filling phase of the injection molding process in RHCM. PS, ABS, PP, ABS/polymethylmethacrylate (ABS/PMMA), reinforced polypropylene glass fiber (FRPP), and ABS/PMMA/nano-C_a_CO_3_ were the materials used. The results showed that the tensile strength of PS, ABS, ABS/PMMA, ABS/PMMA/nano-C_a_CO_3_, and FRPP in the molded samples with or without weld lines gradually decreased, while the tensile strength of PP was restricted in a small range. Random fluctuations increased with the increase in T_CS_. For the formed samples with weld lines, with the rise in T_CS_, the tensile strengths of PS and PP gradually increased, while the tensile strengths of ABS, ABS/PMMA, ABS/PMMA/nano-C_a_CO_3_, and FRPP gradually decreased. Since its T_CS_ is higher than 100 °C, the tensile strength of FRPP with weld lines reduced gradually as T_CS_ increased.

Zhao et al. [127] investigated the efficiency of the electric heating technology applied during the RHCM process to heat the mold surface. Two different mold structures were designed and compared with and without a detached cooling plate for a large LCD TV panel. As a result, the mold’s heating efficiency with a separate cooling plate was better than that of the mold without a cooling plate. It was discovered that the panel molded using the CIM process had visible weld marks as a result of the gating system’s multiple gates and a low surface gloss. The electric heating RHCM process completely eliminated the weld marks on the surface of the panel and greatly improved the surface gloss. The high cavity surface temperature avoided premature melt freezing during filling and packing and improved polymer molecule chain entanglement for better aesthetics and weld strength.

Wang et al. [54] studied the impact of RHCM process on surface appearance and strength on the plastic parts. The cooling channel was placed perpendicular to the cartridge heater to enhance the structural strength and rigidity of the cavity plate, as shown in Figure 18. The results showed that the weld lines on the surface of RHCM parts vanished entirely compared to CIM. The jetting marks were also completely eliminated on RHCM part surface. Thus, compared to CIM, RHCM could greatly improve surface gloss, especially for fiber-reinforced plastics. Furthermore, compared to CIM, RHCM reduced the part’s tensile strength. RHCM decreased the tensile strength of ABS/PMMA plastics by weld lines. RHCM increased the tensile strength of the parts with weld lines in fiber-reinforced PP. The RHCM process could increase the weld line factor of ABS/PMMA and 20% glass fiber-reinforced PP.

In conclusion, thermal assistance through the use of RHCM is an alternative to increasing the quality of the product by improving its mechanical and aesthetic properties. In order to obtain reliable and optimum results, more research on this technique needs to be conducted. For mold manufacturing and injection processes, complex mold configurations involve an additional cost. However, there are still limited studies available on molds fabricated using RT in RHCM to better understand the mechanical properties and aesthetic properties (surface appearance) of the molded part, as most studies focused on the use of conventional molds in the RHCM process.

## 8. Summary and Future Works

It can be concluded from the completed review that the RT technique has not been well established and implemented in RHCM technology. RT provides a faster manufacturing speed, which lowers costs and saves project time, both of which are critical for successfully completing testing and moving into full production. The review on various materials in fabricating mold inserts in RT was studied in terms of several aspects (flexural strength, hardness, thermal conductivity, tensile strength, compressive strength, density, thermal diffusivity, and surface roughness). The study found that, by using appropriate materials, the essential properties of mold insert materials were improved, including excellent machinability and high compressive strength, combined with sufficient toughness, good resistance to heat and wear, and high thermal conductivity. Furthermore, the review highlighted the ability of filler particles in epoxy resin to improve the properties of epoxy resin mold materials according to the application requirements (such as compressive strength, density, thermal diffusivity, and thermal conductivity).

Moreover, the effectiveness of RHCM in removing weld lines and improving cosmetic appearance of the product was discussed, and the efficiencies of conventional mold inserts in RHCM were also presented. This review highlighted the efficiency of RHCM with the application of different heating technologies in reducing weld lines and enhancing the strength of the molded parts produced. However, most of the research only focused on using conventional mold material as mold insert cavities, and there remains a lack of support for and verification of the application of RT technique in fabricating mold inserts involving a lower cost than conventional material molds. Considering the gaps in the literature, this review proposes the following future work:Previous researchers attempted to integrate new compositions of mold insert materials such as epoxy resin composites containing aluminum (Al) particles as filler to improve the mechanical properties of the injected parts. Thus, various types of metal fillers that can be used as fillers in epoxy resin composites should be studied. Studies should be aimed at looking for a new metal epoxy composites as the mold insert materials in RT with the best compositions of epoxy resin ratio and metal filler ratio.The performance of new metal epoxy composite designs in terms of flexural strength, hardness, thermal conductivity, tensile strength, compressive strength, density, thermal diffusivity, and surface roughness must be studied before they can be used as mold inserts in the injection molding process.Previous research on RHCM parts produced with conventional mold inserts discovered that the weld line was reduced while the strength was increased. Thus, there is a lot of potential in researching the use of mold inserts made by RT in RHCM technology to yield molded parts with good surface appearance and properties.Previous research did not yield a positive correlation between the heating time by induction heating and temperature distribution pattern on mold plates produced by the RT technique. Therefore, further investigation is required.Lastly, investigating the temperature distribution on the surface of the mold insert is critical in determining the effect of heating on the molded parts. Thus, further studies into the temperature distribution on the mold insert surface of different materials of mold inserts in RHCM should be conducted.

## Figures and Tables

**Figure 1 materials-15-03725-f001:**
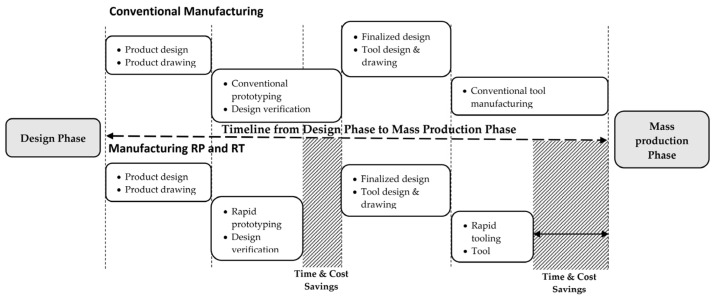
Summary of RT integration in manufacturing.

**Figure 2 materials-15-03725-f002:**
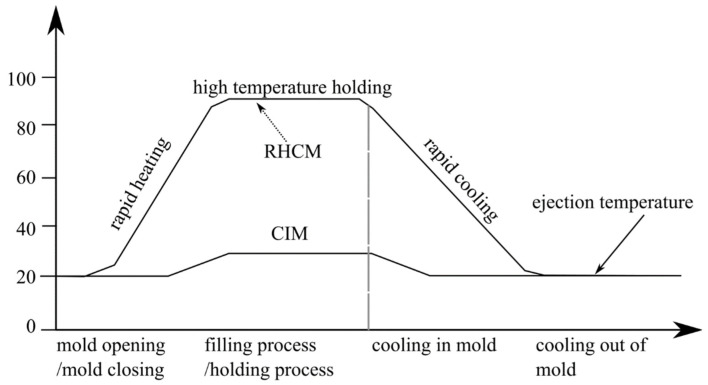
The cycle in the RHCM process [88].

**Figure 3 materials-15-03725-f003:**
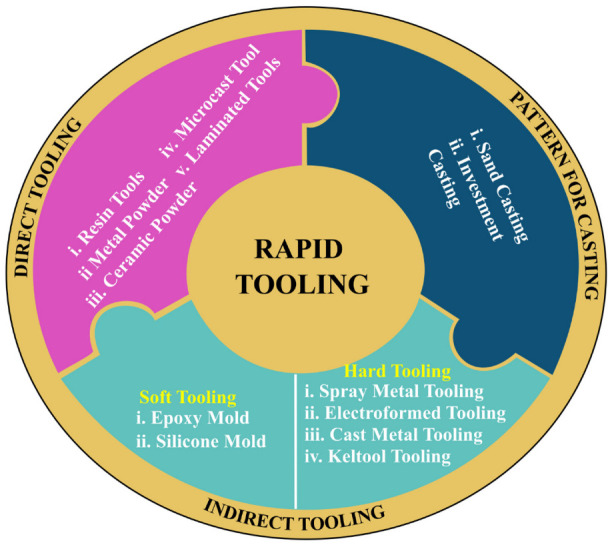
Classification of RT.

**Figure 4 materials-15-03725-f004:**
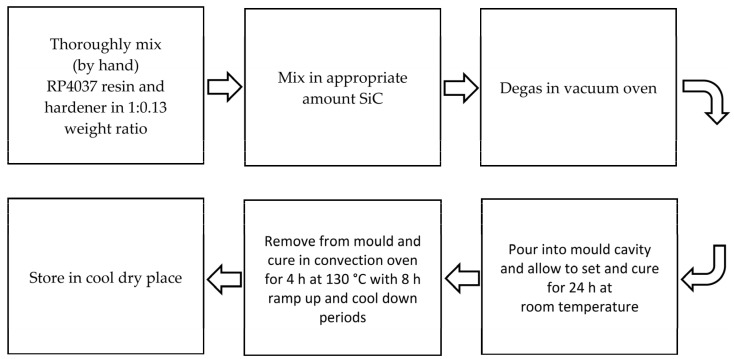
Procedure for tooling board preparation [136].

**Figure 5 materials-15-03725-f005:**
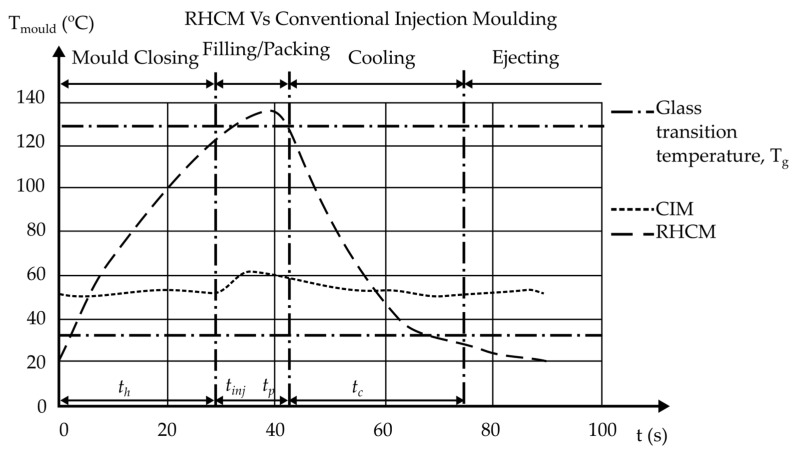
Example of mold temperature profile in RHCM [128].

**Figure 6 materials-15-03725-f006:**
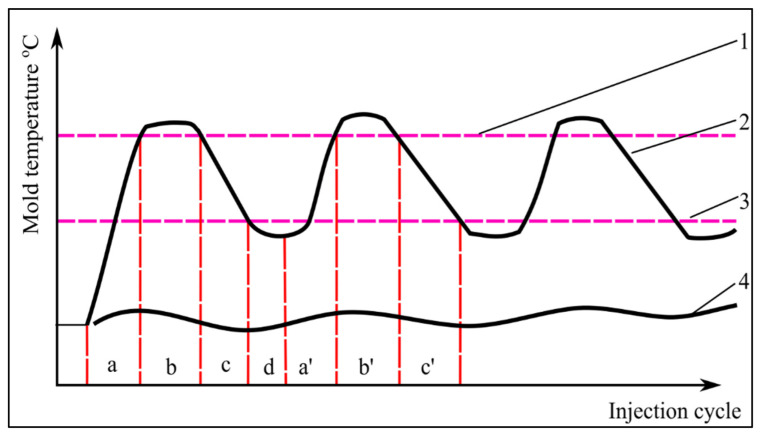
Mold temperature of RHCM technology and conventional injection method [56].

**Figure 7 materials-15-03725-f007:**
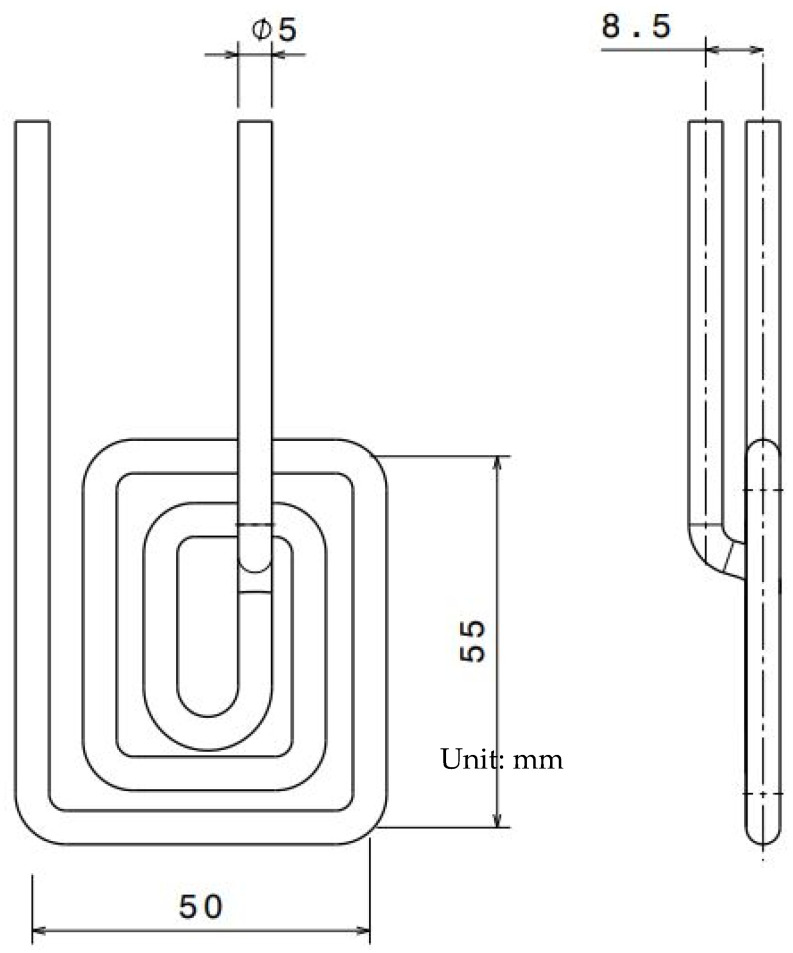
Induction coil’s geometry and dimensions.

**Figure 8 materials-15-03725-f008:**
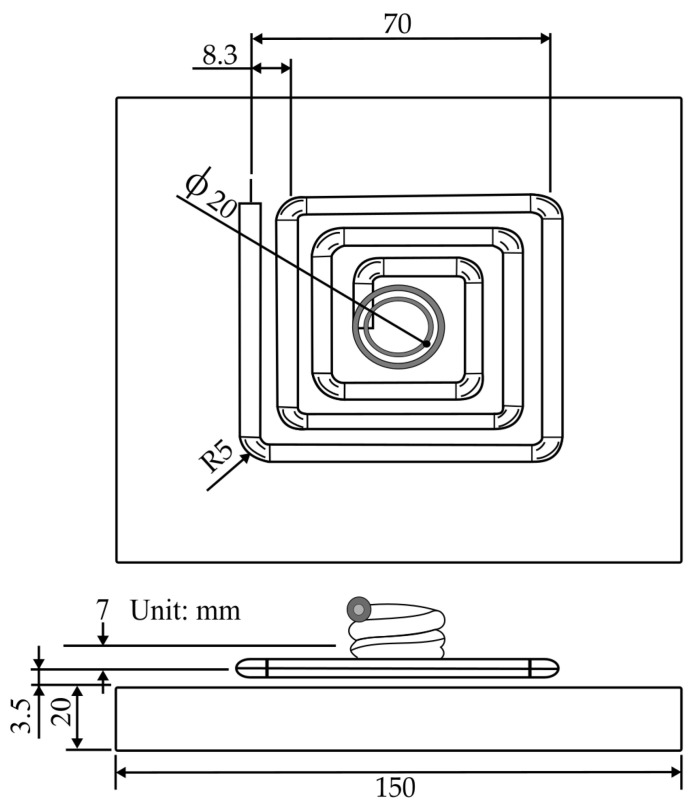
The geometry and dimensions of induction coil.

**Figure 9 materials-15-03725-f009:**
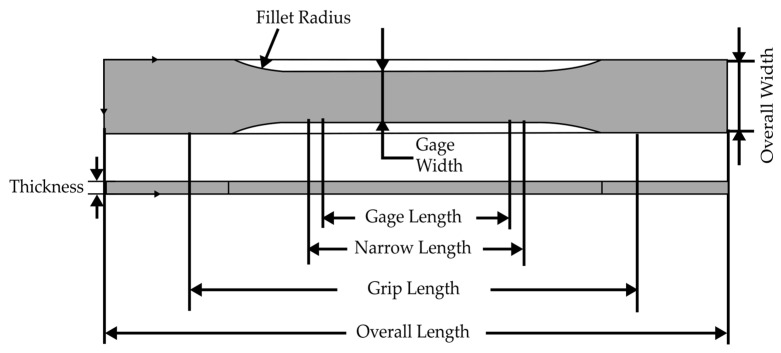
Specimen of tensile testing [149].

**Figure 10 materials-15-03725-f010:**
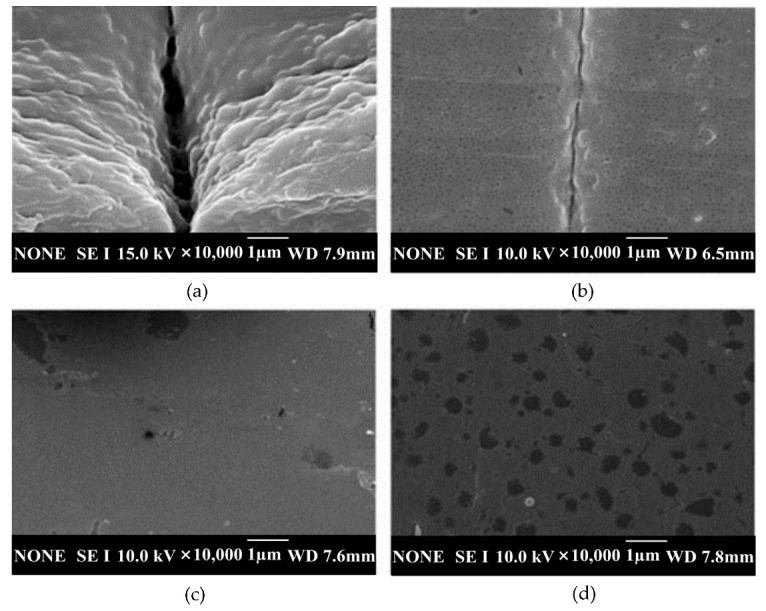
SEM images of one gate and two opposite gates: (**a**) two opposite gates at the temperature of 75 °C at point O without vapor chamber system; (**b**) two opposite gates at the temperature of 75 °C at point O with vapor chamber system; (**c**) two opposite gates at the temperature of 110 °C at point O with vapor chamber system; (**d**) one gate at the temperature of 75 °C at point O without vapor chamber system [46].

**Figure 11 materials-15-03725-f011:**
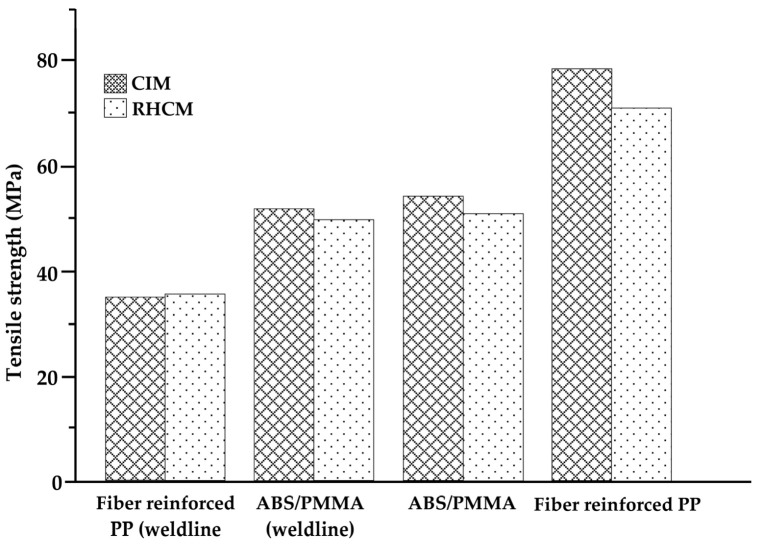
The tensile strength with and without the weld line of the CIM and RHCM parts.

**Figure 12 materials-15-03725-f012:**
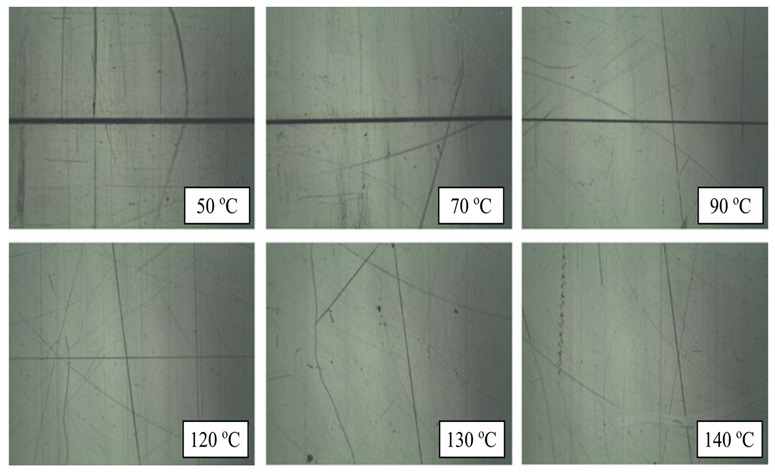
Micrograph of the weld marks or weld lines on the surface of the molded specimen under different cavity surface temperatures just before filling [47].

**Figure 13 materials-15-03725-f013:**
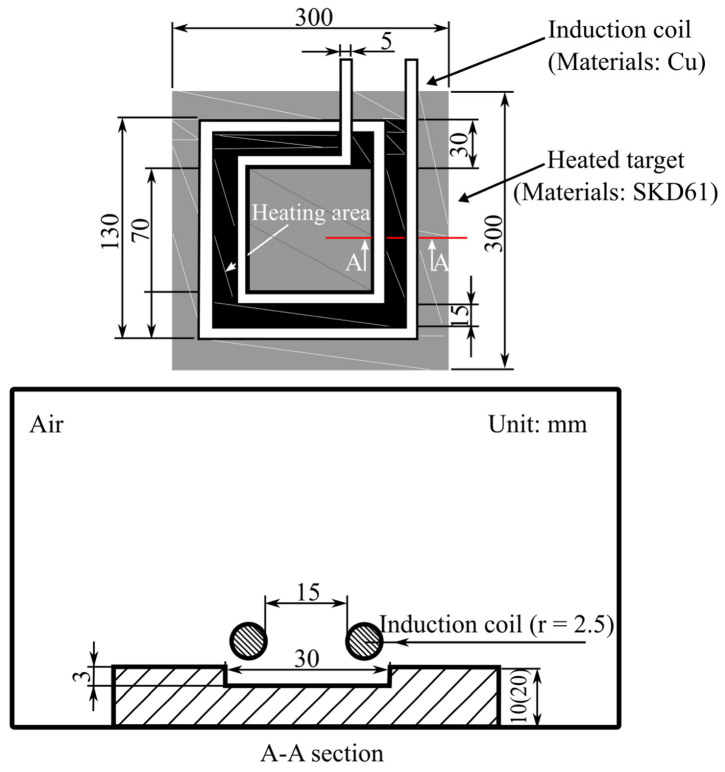
Physical model of induction heating [45].

**Figure 14 materials-15-03725-f014:**
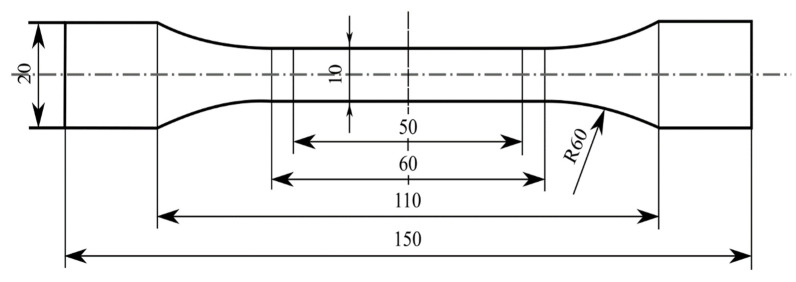
The dimensions of the dumbbell-shaped specimens [48].

**Figure 15 materials-15-03725-f015:**
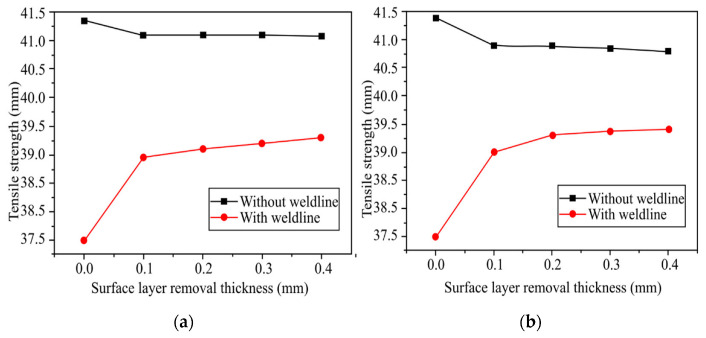
Tensile strength of RHCM parts: (**a**) when the surface layer was removed from the surface adjacent to the heated stationary half, and (**b**) when the surface layer was removed from the surface adjacent to the moving half of the surface [48].

**Figure 16 materials-15-03725-f016:**
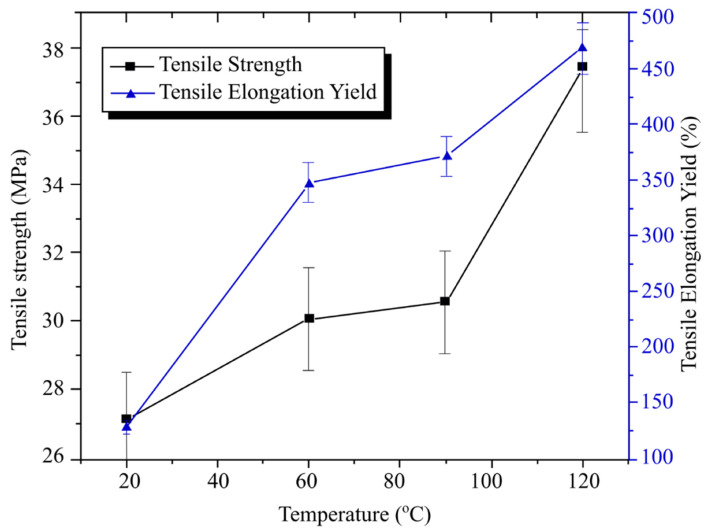
The mechanical properties of samples under different mold surface temperatures [49].

**Figure 17 materials-15-03725-f017:**
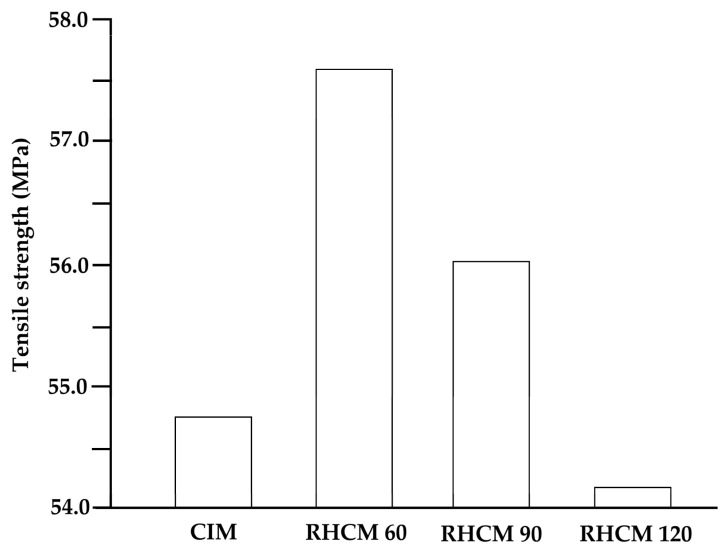
Tensile strength of RHCM and CIM samples.

**Figure 18 materials-15-03725-f018:**
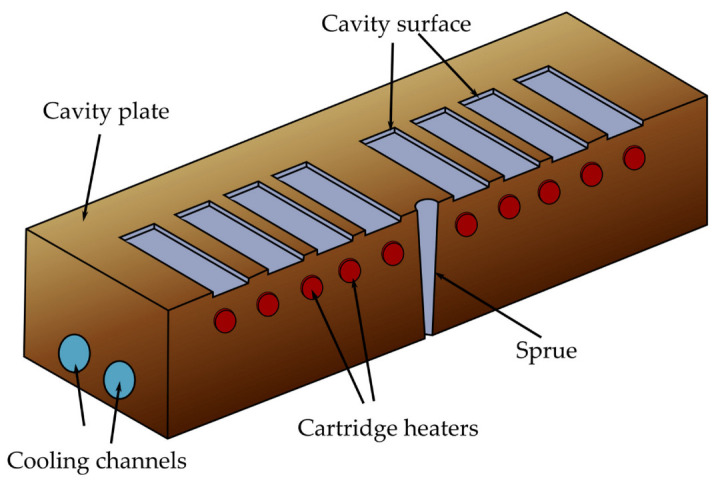
Location in RHCM for heaters and cooling line.

**Table 1 materials-15-03725-t001:** RCHM Technologies.

No.	RHCM Technology	Researchers
1	Electromagnetic induction heating	Chen et al., 2006 [39]
2	Induction heating	Huang and Tai, 2009 [43]; Huang et al., 2010 [44]; Nian et al., 2014 [45]
3	Hot oil	Huang and Tai, 2009 [43]
4	Combination of hot oil and induction heating	Huang and Tai, 2009 [43]
5	Steam heating (vapor chamber)	Tsai, 2011 [46]; Wang, 2014 [47]
6	Electric heating	Li, 2016 [48]; Xie, 2017 [49]; Liu, 2020 [50]

**Table 2 materials-15-03725-t002:** Physical and mechanical properties of mold base materials.

Type of Material	Density (g/cm^3^)	Specific Heat (J/g·°C)	Thermal Conductivity (W/m·K)	Hardness (HB)	Authors
Aluminum (2000) series	2.78	0.869	139	135	Ozcelik et al. [108]
AISI (1020) steel	7.87	0.486	51.9	170	Ozcelik et al. [108]
Aluminum (QC-10)	2.85	N/A	159.12	150–170	Raus et al. [113]
Copper Alloy (C18000)	8.81	N/A	225	94	Raus et al. [113]
Steel (PC-20)	7.87	N/A	34.59	264–331	Raus et al. [113]
Carbon steel AISI 1050	7.85	0.486	49.8	196	Tang et al. [114]

**Table 3 materials-15-03725-t003:** Physical and mechanical properties of mold insert materials for CIM.

Type of Material	Density (g/cm^3^)	Specific Heat (J/g·°C)	Thermal Conductivity (W/m·K)	Authors
P20 mold steel	7.8	460	29	Nasir et al. [122]
AISI P20	7.85	460	34	Xiao and Huang [123]
P20 tool steel	7.86	N/A	41.5	Sateesh [124]
Tool steel SKD-61	7.76	460	25	Chiang and Chang [121]
NAK80 (pre-hardened steel)	7.72	N/A	41.3	Chung [125]
(45–55 HRC, STAVAX) (pre-hardened steel)	7.8	460	16	Okubo et al. [126]

**Table 4 materials-15-03725-t004:** Physical and mechanical properties of mold insert materials for RHCM.

Type of Material	Density (g/cm^3^)	Specific Heat (J/g·°C)	Thermal Conductivity (W/m·K)	Authors
Hot work tool steel (Vidar Superior)	7.78	460	30	Li et al. [55]
AISI P20	7.85	460	34	Wang et al. [87]
AMPCO 940	8.71	380	208	Wang at al. [87]
CENA 1	7.78	495	22.9	Wang et al. [47]

**Table 5 materials-15-03725-t005:** Research on mold inserts in RAPID TOOLING.

	Researcher	Epoxy Resin/Hardener	Particles/Fillers Used	Weight Percentage of Filler (wt.%)	Particle Size	Mechanical Test											
						Arithmetic Mean Roughness (Ra) (µm)	Flexural Strength (MPa)	Hardness Test (R_H_)	Thermal Conductivity (W/m·K)	Fatigue Test	Tensile Strength (MPa)	Compressive Strength (MPa)	Vickers Hardness, (kgF/mm^2^)	Shore D Hardness Test	Density (g/cm^3^)	Thermal Diffusivity (mm^2^/s)	Surface Roughness
1.	Tomori et al. (2004) [136]	RP4037 (resin) RP4037 (hardener)	SiC	28.5 34.7 39.9	N/A	1.03 to 1.35	58.75 to 66.49	N/A	N/A	N/A	N/A	N/A	N/A	N/A	N/A	N/A	N/A
2.	Senthilkumar et al. (2012) [137]	Araldite LY 556 (resin)	Al	40 45 50 55 60	45– 150 µm	N/A	N/A	69 to 89	3.97 to 5.39	15,786 to 734	N/A	N/A	N/A	N/A	N/A	N/A	N/A
3.	Srivastava and Verma (2015) [34]	PL-411 (resin) PH-861 (hardener)	Cu Al	1 5 8 10	N/A	N/A	N/A	N/A	N/A	N/A	<85 (pure epoxy)	Cu = 65 at 10 wt.%	Cu = 22.4 at 8 wt.%.	N/A	N/A	N/A	N/A
4.	Fernandes et al. (2016) [35]	RenCast 436 (resin with Al filler) Ren HY 150 (hardener)	Al	21.4	N/A	N/A	N/A	N/A	N/A	N/A	Steel AISI P20 inserts = 20.0 ± 4.5 Epoxy resin/Al inserts = 22.0 ± 5.0	N/A	N/A	Steel AISI P20 inserts = 66 ± 3.2 Epoxy resin/Al inserts = 61 ± 1.6	N/A	N/A	N/A
5	Khushairi et al. (2017) [138]	RenCast CW 47 (resin with Al filler) Ren HY 33 (hardener)	Brass Cu	10 20 30	N/A	N/A	N/A	N/A	Brass: 10% = 1.18, 20% = 1.21, 30% = 1.37 Cu: 10% = 1.66, 20% = 1.73, 30% = 1.87	N/A	N/A	Brass: 10% = 95.61, 20% = 93.23, 30% = 92.69 Cu: 10% = 80.83, 20% = 81.51, 30% = 73.17	N/A	N/A	Brass: 10% = 1.85, 20% = 2.01, 30% = 2.22 Cu: 10% = 1.83, 20% = 1.96, 30% = 2.08	Brass: 10% = 0.644, 20% = 0.657, 30% = 0.740 Cu: 10% = 0.837, 20% = 0.923, 30% = 1.112	N/A
6	Kuo and Lin (2019) [139]	TE-375 (Al filled epoxy resin)	N/A	N/A	N/A	N/A	N/A	N/A	N/A	N/A	N/A	N/A	N/A	N/A	N/A	N/A	Average microgroove depth of Al-filled epoxy resin was 90.5% Average microgroove width of Al-filled epoxy resin was 98.9%

**Table 6 materials-15-03725-t006:** Summary of research on RHCM technologies.

No.	Researcher	Plastic Material Used	Parameter Settings	Output Response	Technology Used	Material for Mold Inserts	Type of Analysis		Result
							Simulation	Experiment	
1	Chen et al., 2006 [39]	ABS	Heating stage (110–180 °C and 110–200 °C) Cooling stage (180–110 °C and 200–110 °C) Mold temperature (50 °C)	Surface marks Weld line strength	Electromagnetic induction heating	AISI 4130 steel	ANSYS	Yes	Heating times, 3–4 s for mold surface temperature to rise from 110 to 180 °C and 200 °C, along with 21 s for cooling time (return to 110 °C) Eliminated the surface marks of the weld line and enhanced the strength of the related weld line
2	Huang and Tai, 2009 [43]	PMMA	Mold temperature (25 °C) Cooling time (20 s) Melt temperature (260 °C)	Mold surface temperature Replication heights of LGP’s microstructures Residual stress in LGP	N/A	Not specified	No	Yes	Induction heating the mold surface to 110 °C could increase the replication rate of the microstructure’s height by up to 95% There was no residual stress in the LGP produced by induction heating
			Mold temperature (80 °C) Cooling time (20 s) Melt temperature (260 °C)		Hot oil		No	Yes	
			Mold temperature (110 °C) Cooling time (20 s) Melt temperature (260 °C)		Combination of hot oil and induction heating		No	Yes	
			Mold temperature (110, 130, and 150 °C) Cooling time (20 s) Melt temperature (260 °C)		Induction heating		No	Yes	
3	Huang et al., 2010 [44]	PMMA	Injection speed (180–200 mm/s) Packing pressure (1st stage 50–70 Mpa, 2nd stage 40 Mpa) Packing time (4–8 s) Mold temperature (60–80 °C) Cooling time (30–40 s) Mold surface temperature (110–150 °C)	Power rates Optimum processing parameters Quality of microfeature heights and angles	Induction heating	Ni	No	Yes	Optimum process parameters: injection speed (180 mm/s) packing pressure (70 Mpa), packing time (8 s), mold temperature (70 °C), cooling time (30 s), and mold surface temperature (150 °C). Replication effect on microfeatures was significantly improved by induction heating
4	Tsai, 2011 [46]	ABS	Two gates, cavity temperature = 75 °C, no vapor chamber Two gates, cavity = 75 °C temperature, with vapor chamber Two gates, cavity temperature = 110 °C, with vapor chamber One gate, cavity temperature = 75 °C, no vapor chamber	Tensile strength	Steam heating (vapor chamber)	P20 mould steel	No	Yes	The two gate/vapor chamber system’s tensile strength increased by 3.2% when preheating temperatures increased from 75 to 110 °C
5	Wang, 2013 [54]	ABS/PPMA Fiber-reinforced plastic PP + 20% glass fiber	Heating time (10, 20, 30, 40, 50, and 60 s) Cooling time (20,30,40, 50, and 60 s) High- and low-temperature holding time (10 s)	Weld line Tensile strength	Electric heating (cartridge heater)	AISI H13	ANSYS	Yes	RHCM process could improve the weld line factor for both materials RHCM process reduced the tensile strength of the part without weld line
6	Wang, 2014 [47]	PC	Mold heating time (18, 24, 25, and 36 s) Mold cooling time (25, 32, 38, and 46 s)	Weld line	Steam heating	CENA1	Moldflow Insight ANSYS	Yes	Weld marks on the part surface could be significantly reduced by increasing the cavity surface’s temperature just before filling Surface gloss of product produced by RHCM was more than 90%.
7	Nian et al., 2014 [45]	Not specified	Mold temperature (between 120 and 150 °C) Thicknesses of the heated target (10–20 mm) Pitch of the coil turns (10–20 mm) Heating distance (5–9 mm) Position of the induction coil (0–12 mm) Working frequency (30–40 kHz) Waiting time (2–6 s)	Heating rate Temperature difference	Induction heating	SKD61	COMSOL Multiphysics	Yes	Heating rate was increased by 19.5%, from 3.3 °C/s to 4 °C/s Heating uniformity was increased by 62.9%
8	Li, 2016 [48]	iPP	Heated mold temperature for RHCM (120 °C) Mold temperature for CIM (25 °C) Packing pressure (50 Mpa) Cooling time (30 s)	Weld line Tensile strength	Electric heating (electrical heating rods)	Not specified	N/A	Yes	Weld line decreased tensile strength, but RHCM reduced the weld line’s tensile strength reduction effect.
9	Xie, 2017 [49]	PP	Silicon insert surface temperature (20, 60, 100, and 140 °C) Melt temperature (230 °C) Injection pressure (30 Mpa) Injection speed (60 mm/s) Screw back (20 mm) Sample thickness (0.6 mm)	Weld line	Electric heating (thin-film resistance heater, graphene coating)	Silicon insert (coated with carbide-bonded graphene)	N/A	Yes	Width of weld lines: 16.4 µm at 20 °C, 11.24 µm at 60 °C, and 5.6 µm at 100 °C Weld line disappeared completely at 140 °C
		PS	Silicon insert surface temperature (20, 40, 80, and 100 °C) Melt temperature (200 °C) Injection pressure (30 Mpa) Injection speed (50 mm/s) Screw back (15 mm) Sample thickness (1 mm)	Residual internal stress Replication fidelity				Yes	Residual stress decreased as the surface temperature of silicon insert increased Coating the silicon insert with carbide-bonded graphene could improve replication fidelity
10	Liu, 2020 [50]	PP with 30% short glass fiber	Melt temperature (240 °C) Injection pressure (60 Mpa) Injection velocity (45%) Packing time (9 s) Packing pressure (50 Mpa) Mold heating temperature (60/90/120 °C) Cooling time (30 s)	Microstructure Tensile properties Surface quality	Electric heating (electrical heating rods)	Not specified	Autodesk Moldflow	Yes	Tensile strength of RHCM parts reached peak at 60 °C mold heating temperature The sample’s surface gloss increased as the mold cavity surface temperature increased, but decreased as the mold heating temperature increased above 90 °C

**Table 7 materials-15-03725-t007:** Research on improved weld line strength performance and surface quality using RHCM.

No.	Authors	Plastic Material	Heating Technology	Parameters	Response	Results
1	Wang et al. [37]	PS PP ABS ABS/PMMA ABS/PMMA/nano-C_a_CO_3_ FRPP	Electric heating	Cavity surface temperature, Tcs	Tensile strength	The tensile strength for molded specimens without weld lines except for PP was decreased slightly and gradually
2	Zhao et al. [127]	ABS	Electric heating	Designed of RHCM mold structures	Weld marks	The LCD TV panel did not have any surface defects such as flow marks and weld marks
3	Wang et al. [54]	ABS/PPMA Fiber-reinforced plastic, PP + 20% glass fiber	Electric heating	Thermal responses	Weld lines Tensile strength	The RHCM process could greatly increase the surface gloss of the part, especially for the fiber-reinforced plastics. The RHCM process reduced the tensile strength of the part without weld line

## Data Availability

Not applicable.

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
