# Peer review of "Potential of Rapid Tooling in Rapid Heat Cycle Molding: A Review"

_materials, 2022, doi:10.3390/ma15103725_

Round 1
Reviewer 1 Report
Title: Potential of Rapid Tooling in Rapid Heat Cycle Moulding (RHCM): A Review
* The abbreviation RHCM should be eliminated from the title.
* Abstract should underscore the difference between rapid tooling and rapid heating cycle moulding. The abstract can mention the reason for the improvement in the surface quality by rapid heating cycle moulding.
* The introduction may be structured to present the continuing development in rapid heating cycle moulding. Figure 1 caption can be clear. Table 5 may be inserted in the introduction. The objective of the review should also be detailed.
* Please mention the application of mould base material and mould insert materials.
* Please write about the effect of processing of moulding on the dimension and the thermal conductivity.
* Figure 16 should be replotted without the straight lines joining two points. Please explain tensile elongation yield in Figure 21. What is the effect of surface finish on the tensile strength and elongation of GFRPP composites?
* The gap in the literature may be lengthened in the section 8.
Author Response
Thanks for your valuable comments. We really appreciate it. Please refer to the attached file.

Reviewer 2 Report
The article is a review on Rapid Tooling in Rapid Heat Cycle Moulding.
The contents are clear and well expressed. The work presents useful references for research work.
Author Response

(The authors gave the same response as above.)

Reviewer 3 Report
The aim and scope of this paper are sound and worthy to be pursued. But some of the work is incomplete. I suggest accepting the paper if the authors succeed in addressing these comments.
1) In the Figure9, the cycle time is not described. Are the cycle time the same for the same part using RHCM and conventional injection method?
2) There are many ways of electric heating. Should the introduction of electric heating and the description in some tables be more subdivided instead of theelectric heating.
3) The introduction of rapid heating is not comprehensive enough.For example, the RHCM method can be used to make different areas of the same part with different properties.
DOI:10.1515/polyeng-2017-0100
4) Section 6 and 7 cover many repetitive tasks, such as Weld lines.
5) In Section 6, a brief introduction to the current statusand products of the company can be given.
6) L190 mentioned the importance of rapid cooling.However, this paper does not give a detailed introduction. As for the RHCM technology in the title, the paper only involves rapid heating without rapid cooling, so relevant parts must be added.
7) In this paper, RHCM and CIM are mainly compared,but the application of RHCM in μIM is not mentioned at all, which should be added appropriately. Therefore, the insert manufacturing and materials need to be improved. If not, the authors should consider changing the title.
Author Response

(The authors gave the same response as above.)
